# DORA THE EXPLORER: DIRECTED OUTREACHING REINFORCEMENT ACTION-SELECTION

**Leshem Choshen**[*]
School of Computer Science and Engineering
and Department of Cognitive Sciences
The Hebrew University of Jerusalem
leshem.choshen@mail.huji.ac.il

**Lior Fox**[*]
The Edmond and Lily Safra Center for Brain Sciences
The Hebrew University of Jerusalem
lior.fox@mail.huji.ac.il

**Yonatan Loewenstein**
The Edmond and Lily Safra Center for Brain Sciences,
Departments of Neurobiology and Cognitive Sciences
and the Federmann Center for the Study of Rationality
The Hebrew University of Jerusalem
yonatan@huji.ac.il

## ABSTRACT

Exploration is a fundamental aspect of Reinforcement Learning, typically implemented using stochastic action-selection. Exploration, however, can be more efficient if directed toward gaining new world knowledge. Visit-counters have been proven useful both in practice and in theory for directed exploration. However, a major limitation of counters is their locality. While there are a few model-based solutions to this shortcoming, a model-free approach is still missing. We propose $E$-values, a generalization of counters that can be used to evaluate the propagating exploratory value over state-action trajectories. We compare our approach to commonly used RL techniques, and show that using $E$-values improves learning and performance over traditional counters. We also show how our method can be implemented with function approximation to efficiently learn continuous MDPs. We demonstrate this by showing that our approach surpasses state of the art performance in the Freeway Atari 2600 game.

## 1 INTRODUCTION

> *"If there's a place you gotta go - I'm the one you need to know."*
>
> (Map, Dora The Explorer)

We consider Reinforcement Learning in a Markov Decision Process (MDP). An MDP is a five-tuple $M = (\mathcal{S}, \mathcal{A}, P, R, \gamma)$ where $\mathcal{S}$ is a set of *states* and $\mathcal{A}$ is a set of *actions*. The dynamics of the process is given by $P(s'|s, a)$ which denotes the *transition probability* from state $s$ to state $s'$ following action $a$. Each such transition also has a distribution $R(r|s, a)$ from which the *reward* for such transitions is sampled. Given a *policy* $\pi : \mathcal{S} \to \mathcal{A}$, a function – possibly stochastic – deciding which actions to take in each of the states, the state-action value function $Q^\pi : \mathcal{S} \times \mathcal{A} \to \mathbb{R}$ satisfies:

$$Q^\pi(s, a) = \mathop{\mathbb{E}}_{r,s' \sim R \times P(\cdot|s,a)} [r + \gamma Q^\pi(s', \pi(s'))]$$

where $\gamma$ is the *discount factor*. The agent's goal is to find an optimal policy $\pi^*$ that maximizes $Q^\pi(s, \pi(s))$. For brevity, $Q^{\pi^*} \triangleq Q^*$. There are two main approaches for learning $\pi^*$. The first is a *model-based* approach, where the agent learns an internal model of the MDP (namely $P$ and $R$). Given a model, the optimal policy could be found using dynamic programming methods such as Value Iteration (Sutton & Barto, 1998). The alternative is a *model-free* approach, where the agent learns only the value function of states or state-action pairs, without learning a model (Kaelbling et al., 1996)[1].

---

[*]These authors contributed equally to this work

[1]Supplementary code for this paper can be found at https://github.com/borgr/DORA/

The ideas put forward in this paper are relevant to any model-free learning of MDPs. For concreteness, we focus on a particular example, $Q$-Learning (Watkins & Dayan, 1992; Sutton & Barto, 1998). $Q$-Learning is a common method for learning $Q^*$, where the agent iteratively updates its values of $Q(s, a)$ by performing actions and observing their outcomes. At each step the agent takes action $a_t$ then it is transferred from $s_t$ to $s_{t+1}$ and observe reward $r$. Then it applies the update rule regulated by a *learning rate* $\alpha$:

$$Q(s_t, a_t) \leftarrow (1 - \alpha) Q(s_t, a_t) + \alpha \left( r + \gamma \max_a Q(s_{t+1}, a) \right).$$

## 1.1 EXPLORATION AND EXPLOITATION

Balancing between *Exploration* and *Exploitation* is a major challenge in Reinforcement Learning. Seemingly, the agent may want to choose the alternative associated with the highest expected reward, a behavior known as exploitation. However, in that case it may fail to learn that there are better options. Therefore exploration, namely the taking of new actions and the visit of new states, may also be beneficial. It is important to note that exploitation is also inherently relevant for learning, as we want the agent to have better estimations of the values of valuable state-actions and we care less about the exact values of actions that the agent already knows to be clearly inferior.

Formally, to guarantee convergence to $Q^*$, the Q-Learning algorithm must visit each state-action pair infinitely many times. A naive random walk exploration is sufficient for converging asymptotically. However, such random exploration has two major limitations when the learning process is finite. First, the agent would not utilize its current knowledge about the world to guide its exploration. For example, an action with a known disastrous outcome will be explored over and over again. Second, the agent would not be biased in favor of exploring unvisited trajectories more than the visited ones – hence "wasting" exploration resources on actions and trajectories which are already well known to it.

A widely used method for dealing with the first problem is the $\epsilon$-greedy schema (Sutton & Barto, 1998), in which with probability $1 - \epsilon$ the agent greedily chooses the best action (according to current estimation), and with probability $\epsilon$ it chooses a random action. Another popular alternative, emphasizing the preference to learn about actions associated with higher rewards, is to draw actions from a *Boltzmann Distribution* (*Softmax*) over the learned $Q$ values, regulated by a *Temperature* parameter. While such approaches lead to more informed exploration that is based on learning experience, they still fail to address the second issue, namely they are not **directed** (Thrun, 1992) towards gaining more knowledge, not biasing actions in the direction of unexplored trajectories.

Another important approach in the study of efficient exploration is based on *Sample Complexity of Exploration* as defined in the PAC-MDP literature (Kakade et al., 2003). Relevant to our work is Delayed Q Learning (Strehl et al., 2006), a model-free algorithm that has theoretical PAC-MDP guarantees. However, to ensure these theoretical guarantees this algorithm uses a conservative exploration which might be impractical (see also (Kolter & Ng, 2009) and Appendix B).

## 1.2 CURRENT DIRECTED EXPLORATION AND ITS LIMITATIONS

In order to achieve directed exploration, the estimation of an exploration value of the different state-actions (often termed *exploration bonus*) is needed. The most commonly used exploration bonus is based on **counting** (Thrun, 1992) – for each pair $(s, a)$, store a counter $C(s, a)$ that indicates how many times the agent performed action $a$ at state $s$ so far. Counter-based methods are widely used both in practice and in theory (Kolter & Ng, 2009; Strehl & Littman, 2008; Guez et al., 2012; Busoniu et al., 2008). Other options for evaluating exploration include **recency** and **value difference** (or **error**) measures (Thrun, 1992; Tokic & Palm, 2011). While all of these exploration measures can be used for directed exploration, their major limitation in a *model-free* settings is that the exploratory value of a state-action pair is evaluated with respect only to its immediate outcome, one step ahead. It seems desirable to determine the exploratory value of an action not only by how much new immediate knowledge the agent gains from it, but also by how much more new knowledge *could* be gained from a trajectory starting with it. The goal of this work is to develop a measure for such exploratory values of state-action pairs, in a *model-free* settings.

## 2 Learning Exploration Values

### 2.1 Propagating Exploration Values

The challenge discussed in 1.2 is in fact similar to that of learning the value functions. The value of a state-action represents not only the immediate reward, but also the temporally discounted sum of expected rewards over a trajectory starting from this state and action. Similarly, the "exploration-value" of a state-action should represent not only the immediate knowledge gained but also the expected future gained knowledge. This suggests that a similar approach to that used for value-learning might be appropriate for learning the exploration values as well, using exploration bonus as the immediate reward. However, because it is reasonable to require exploration bonus to decrease over repetitions of the same trajectories, a naive implementation would violate the Markovian property.

This challenge has been addressed in a *model-based* setting: The idea is to use at every step the current estimate of the parameters of the MDP in order to compute, using dynamic programming, the future exploration bonus (Little & Sommer, 2014). However, this solution cannot be implemented in a *model-free* setting. Therefore, a satisfying approach for propagating directed exploration in *model-free* reinforcement learning is still missing. In this section, we propose such an approach.

### 2.2 $E$-Values

We propose a novel approach for directed exploration, based on two parallel MDPs. One MDP is the original MDP, which is used to estimate the value function. The second MDP is identical except for one important difference. We posit that there are no rewards associated with any of the state-actions. Thus, the true value of all state-action pairs is $0$. We will use an RL algorithm to "learn" the "action-values" in this new MDP which we denote as $E$-values. We will show that these $E$-values represent the missing knowledge and thus can be used for propagating directed exploration. This will be done by initializing $E$-values to $1$. These positive initial conditions will subsequently result in an optimistic bias that will lead to directed exploration, by giving high estimations only to state-action pairs from which an optimistic outcome has not yet been excluded by the agent's experience.

Formally, given an MDP $M = (\mathcal{S}, \mathcal{A}, P, R, \gamma)$ we construct a new MDP $M' = (\mathcal{S}, \mathcal{A}, P, \mathbf{0}, \gamma_E)$ with $\mathbf{0}$ denoting the identically zero function, and $0 \leq \gamma_E < 1$ is a discount parameter. The agent now learns both $Q$ and $E$ values concurrently, while initially $E(s, a) = 1$ for all $s, a$. Clearly, $E^* = \mathbf{0}$. However intuitively, the value of $E(s, a)$ at a given timestep during training stands for the knowledge, or uncertainty, that the agent has regarding this state-action pair. Eventually, after enough exploration, there is no additional knowledge left to discover which corresponds to $E(s, a) \rightarrow E^*(s, a) = 0$.

For learning $E$, we use the SARSA algorithm (Rummery & Niranjan, 1994; Sutton & Barto, 1998) which differs from Watkin's $Q$-Learning by being *on-policy*, following the update rule:
$$E(s_t, a_t) \leftarrow (1 - \alpha_E) E(s_t, a_t) + \alpha_E (r + \gamma_E E(s_{t+1}, a_{t+1}))$$
Where $\alpha_E$ is the learning rate. For simplicity, we will assume throughout the paper that $\alpha_E = \alpha$.

Note that this learning rule updates the $E$-values based on $E(s_{t+1}, a_{t+1})$ rather than $\max_a E(s_{t+1}, a)$, thus not considering potentially highly informative actions which are never selected. This is important for guaranteeing that exploration values will decrease when repeating the same trajectory (as we will show below). Maintaining these additional updates doesn't affect the asymptotic space/time complexity of the learning algorithm, since it is simply performing the same updates of a standard $Q$-Learning process twice.

### 2.3 $E$-Values as Generalized Counters

The logarithm of $E$-Values can be thought of as a generalization of visit counters, with propagation of the values along state-action pairs. To see this, let us examine the case of $\gamma_E = 0$ in which there is no propagation from future states. In this case, the update rule is given by:
$$E(s, a) \leftarrow (1 - \alpha) E(s, a) + \alpha (0 + \gamma_E E(s', a')) = (1 - \alpha) E(s, a)$$
So after being visited $n$ times, the value of the state-action pair is $(1 - \alpha)^n$, where $\alpha$ is the learning rate. By taking a logarithm transformation, we can see that $\log_{1-\alpha}(E) = n$. In addition, when $s$ is a terminal state with one action, $\log_{1-\alpha}(E) = n$ for any value of $\gamma_E$.

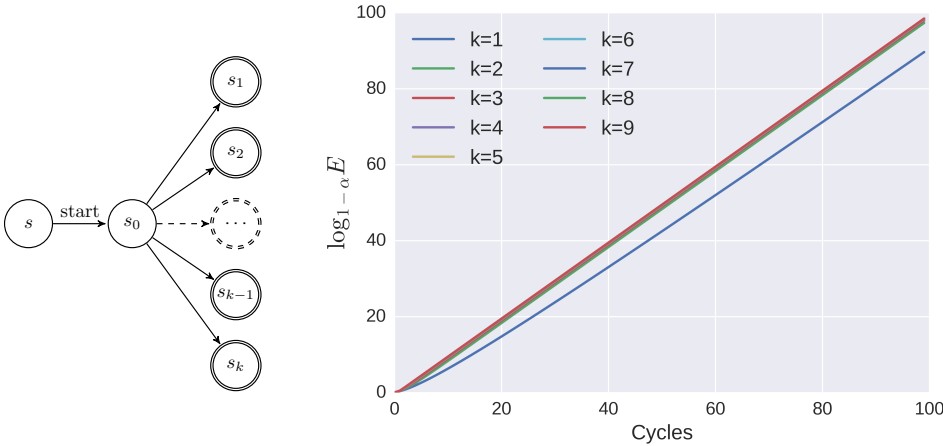

Figure 1: Left: Tree MDP, with $k$ leaves. Tree: $\log_{1-\alpha} E(s, \text{start})$ as function of visit cycles, for different trees of $k$ leaves (color coded). For each $k$, a cycle consists of visiting all leaves, hence $k$ visits of the start action. $\log_{1-\alpha} E$ behaves as a generalized counter, where each cycle contributes approximately one generalized visit.

When $\gamma_E > 0$ and for non-terminal states, $E$ will decrease more slowly and therefore $\log_{1-\alpha} E$ will increase more slowly than a counter. The exact rate will depend on the MDP, the policy and the specific value of $\gamma_E$. Crucially, for state-actions which lead to many potential states, each visit contributes less to the generalized counter, because more visits are required to exhaust the potential outcomes of the action. To gain more insight, consider the MDP depicted in Figure 1 left, a tree with the root as initial state and the leaves as terminal states. If actions are chosen sequentially, one leaf after the other, we expect that each complete round of choices (which will result with $k$ actual visits of the $(s, \text{start})$ pair) will be roughly equivalent to one generalized counter. Simulation of this and other simple MDPs show that $E$-values behave in accordance with such intuitions (see Figure 1 right).

An important property of $E$-values is that they decrease over repetitions. Formally, by completing a trajectory of the form $s_0, a_0, \ldots, s_n, a_n, s_0, a_0$ in the MDP, the maximal value of $E(s_i, a_i)$ will decrease. To see this, assume that $E(s_i, a_i)$ was maximal, and consider its value after the update:

$$E(s_i, a_i) \leftarrow (1 - \alpha) E(s_i, a_i) + \alpha \gamma_E E(s_{i+1}, a_{i+1})$$

Because $\gamma_E < 1$ and $E(s_{i+1}, a_{i+1}) \leq E(s_i, a_i)$, we get that after the update, the value of $E(s_i, a_i)$ decreased. For any non-maximal $(s_j, a_j)$, its value after the update is a convex combination of its previous value and $\gamma_E E(s_k, a_k)$ which is not larger than its composing terms, which in turn are smaller than the maximal $E$-value.

## 3 APPLYING $E$-VALUES

The logarithm of $E$-values can be considered as a generalization of counters. As such, algorithms that utilize counters can be generalized to incorporate $E$-values. Here we consider two such generalizations.

### 3.1 $E$-VALUES AS REWARD EXPLORATION BONUS

In model-based RL, counters have been used to create an augmented reward function. Motivated by this result, augmenting the reward with a counter-based exploration bonus has also been used in model-free RL (Storck et al., 1995; Bellemare et al., 2016). $E$-Values can naturally generalize this approach, by replacing the standard counter with its corresponding generalized counter ($\log_{1-\alpha} E$).

To demonstrate the advantage of using $E$-values over standard counters, we tested an $\epsilon$-greedy agent with an exploration bonus of $\frac{1}{\log_{1-\alpha} E}$ added to the observed reward on the bridge MDP (Figure 2). To measure the learning progress and its convergence, we calculated the mean square error

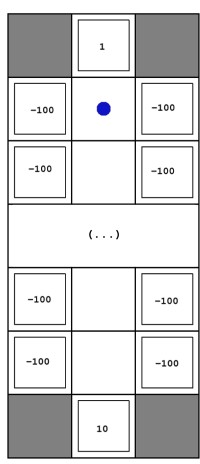

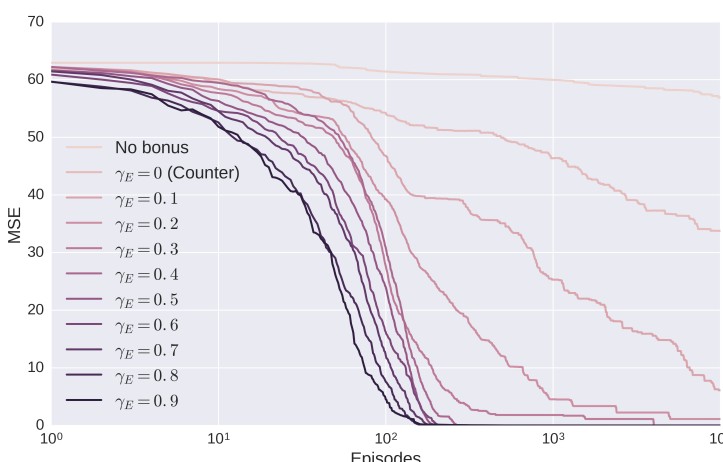

Figure 2: Bridge MDP

Figure 3: MSE between $Q$ and $Q^*$ on optimal policy per episode. Convergence of $\epsilon$-greedy on the short bridge environment ($k = 5$) with and without exploration bonuses added to the reward. Note the logarithmic scale of the abscissa.

$\mathbb{E}_{P(s,a|\pi^*)} \left[ (Q(s,a) - Q^*(s,a))^2 \right]$, where the average is over the probability of state-action pairs when following the optimal policy $\pi^*$. We varied the value of $\gamma_E$ from 0 – resulting effectively in standard counters – to $\gamma_E = 0.9$. Our results (Figure 3) show that adding the exploration bonus to the reward leads to faster learning. Moreover, the larger the value of $\gamma_E$ in this example the faster the learning, demonstrating that generalized counters significantly outperforming standard counters.

## 3.2 $E$-VALUES AND ACTION-SELECTION RULES

Another way in which counters can be used to assist exploration is by adding them to the estimated $Q$-values. In this framework, action-selection is a function not only of the $Q$-values but also of the counters. Several such action-selection rules have been proposed (Thrun, 1992; Meuleau & Bourgine, 1999; Kolter & Ng, 2009). These usually take the form of a deterministic policy that maximizes some combination of the estimated $Q$-value with a counter-based exploration bonus. It is easy to generalize such rules using $E$-values – simply replace the counters $C$ by the generalized counters $\log_{1-\alpha}(E)$.

### 3.2.1 DETERMINIZATION OF STOCHASTIC DECISION RULES

Here, we consider a special family of action-selection rules that are derived as deterministic equivalents of standard stochastic rules. Stochastic action-selection rules are commonly used in RL. In their simple form they include rules such as the $\epsilon$-greedy or Softmax exploration described above. In this framework, exploratory behavior is achieved by stochastic action selection, independent of past choices. At first glance, it might be unclear how $E$-values can contribute or improve such rules. We now turn to show that, by using counters, for every stochastic rule there exist equivalent deterministic rules. Once turned to deterministic counter-based rules, it is again possible improve them using $E$-values.

The stochastic action-selection rules determine the frequency of choosing the different actions in the limit of a large number of repetitions, while abstracting away the specific order of choices. This fact is a key to understanding the relation between deterministic and stochastic rules. An equivalence of two such rules can only be an in-the-limit equivalence, and can be seen as choosing a specific realization of sample from the distribution. Therefore, in order to derive a deterministic equivalent of a given stochastic rule, we only have to make sure that the frequencies of actions selected under both rules are equal in the limit of infinitely many steps. As the probability for each action is likely to depend on the current $Q$-values, we have to consider fixed $Q$-values to define this equivalence.

We prove that given a stochastic action-selection rule $f(a|s)$, every deterministic policy that does not choose an action that was visited too many times until now (with respect to the expected number according to the probability distribution) is a determinization of $f$. Formally, lets assume that given a certain $Q$ function and state $s$ we wish a certain ratio between different choices of actions $a \in A$ to hold. We denote the frequency of this ratio $f_Q(a|s)$. For brevity we assume $s$ and $Q$ are constants and denote $f_Q(a|s) = f(a)$. We also assume a counter $C(s,a)$ is kept denoting the number of choices of $a$ in $s$. For brevity we denote $C(s,a) = C(a)$ and $\sum_a C(s,a) = C$. When we look at the counters after $T$ steps we use subscript $C_T(a)$. Following this notation, note that $C_T = T$.

**Theorem 3.1.** *For any sub-linear function $b(t)$ and for any deterministic policy which chooses at step $T$ an action $a$ such that $\frac{C_T(a)}{T} - f(a) \leq b(t)$ it holds that $\forall a \in \mathcal{A}$*

$$\lim_{T \to \infty} \frac{C_T(a)}{T} = f(a)$$

*Proof.* For a full proof of the theorem see Appendix A in the supplementary materials $\square$

The result above is not a vacuous truth – we now provide two possible determinization rules that achieves it. One rule is straightforward from the theorem, using $b = 0$, choosing $\arg\min_a \frac{C(a)}{C} - f(a)$. Another rule follows the probability ratio between the stochastic policy and the empirical distribution: $\arg\max_a \frac{f(a)}{C(a)}$. We denote this determinization $LLL$, because when generalized counters are used instead of counters it becomes $\arg\max_a \log f(s,a) - \log\log_{1-\alpha} E(s,a)$.

Now we can replace the visit counters $C(s,a)$ with the generalized counters $\log_{1-\alpha}(E(s,a))$ to create Directed Outreaching Reinforcement Action-Selection – DORA the explorer. By this, we can transform any stochastic or counter-based action-selection rule into a deterministic rule in which exploration propagates over the states and the expected trajectories to follow.

**Input:** Stochastic action-selection rule $f$, learning rate $\alpha$, Exploration discount factor $\gamma_E$
initialize $Q(s,a) = 0$, $E(s,a) = 1$;
**foreach** *episode* **do**
    init $s$;
    **while** *not terminated* **do**
        Choose $a = \arg\max_x \log f_Q(x|s) - \log\log_{1-\alpha} E(s,x)$;
        Observe transitions $(s, a, r, s', a')$;
        $Q(s,a) \leftarrow (1-\alpha)Q(s,a) + \alpha(r + \gamma\max_x Q(s',x))$;
        $E(s,a) \leftarrow (1-\alpha)E(s,a) + \alpha\gamma_E E(s',a')$;
    **end**
**end**

**Algorithm 1:** DORA algorithm using $LLL$ determinization for stochastic policy $f$

## 3.3 RESULTS – FINITE MDPS

To test this algorithm, the first set of experiments were done on Bridge environments of various lengths $k$ (Figure 2). We considered the following agents: $\epsilon$-greedy, Softmax and their respective $LLL$ determinizations (as described in 3.2.1) using both counters and $E$-values. In addition, we compared a more standard counter-based agent in the form of a UCB-like algorithm (Auer et al., 2002) following an action-selection rule with exploration bonus of $\sqrt{\frac{\log t}{C}}$. We tested two variants of this algorithm, using ordinary visit counters and $E$-values. Each agent's hyperparameters ($\epsilon$ and temperature) were fitted separately to optimize learning. For stochastic agents, we averaged the results over 50 trials for each execution. Unless stated otherwise, $\gamma_E = 0.9$.

We also used a normalized version of the bridge environment, where all rewards are between 0 and 1, to compare DORA with the Delayed $Q$-Learning algorithm (Strehl et al., 2006).

Our results (Figure 4) demonstrate that $E$-value based agents outperform both their counter-based and their stochastic equivalents on the bridge problem. As shown in Figure 4, Stochastic and counter-based $\epsilon$-greedy agents, as well as the standard UCB fail to converge. $E$-value agents are the first to reach low error values, indicating that they learn faster. Similar results were achieved

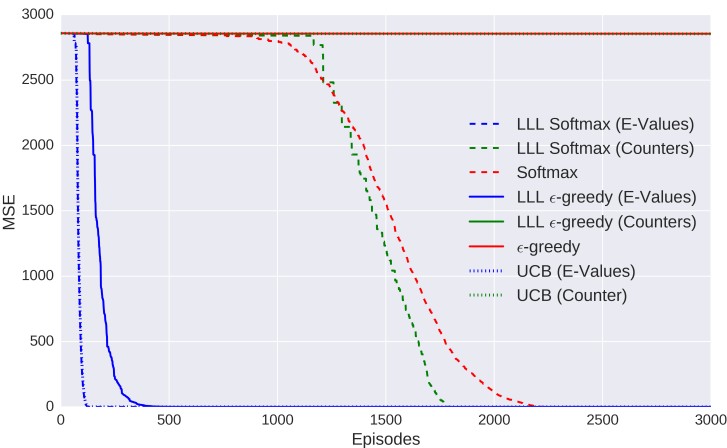

Figure 4: MSE between $Q$ and $Q^*$ on optimal policy per episode. Convergence measure of all agents, long bridge environment ($k = 15$). $E$-values agents are the first to converge, suggesting their superior learning abilities.

on other gridworld environments, such as the Cliff problem (Sutton & Barto, 1998) (not shown). We also achieved competitive results with respect to Delayed $Q$ Learning (see supplementary B and Figure 7 there).

The success of $E$-values based learning relative to counter based learning implies that the use of $E$-values lead to more efficient exploration. If this is indeed the case, we expect $E$-values to better represent the agent's missing knowledge than visit counters during learning. To test this hypothesis we studied the behavior of an $E$-value $LLL$ Softmax on a shorter bridge environment ($k = 5$). For a given state-action pair, a measure of the missing knowledge is the normalized distance between its estimated value ($Q$) and its optimal-policy value ($Q^*$). We recorded $C$, $\log_{1-\alpha}(E)$ and $\left|\frac{Q-Q^*}{Q^*}\right|$ for each $s, a$ at the end of each episode. Generally, this measure of missing knowledge is expected to be a monotonously-decreasing function of the number of visits ($C$). This is indeed true, as depicted in Figure 5 (left). However, considering all state-action pairs, visit counters do not capture well the amount of missing knowledge, as the convergence level depends not only on the counter but also on the identity of the state-action it counts. By contrast, considering the convergence level as a function of the generalized counter (Figure 5, right) reveals a strikingly different pattern. Independently of the state-action identity, the convergence level is a unique function of the generalized counter. These results demonstrate that generalized counters are a useful measure of the amount of missing knowledge.

## 4 $E$-VALUES WITH FUNCTION APPROXIMATION

So far we discussed $E$-values in the tabular case, relying on finite (and small) state and action spaces. However, a main motivation for using model-free approach is that it can be successfully applied in large MDPs where tabular methods are intractable. In this case (in particular for continuous MDPs), achieving directed exploration is a non-trivial task. Because revisiting a state or a state-action pair is unlikely, and because it is intractable to store individual values for all state-action pairs, counter-based methods cannot be directly applied. In fact, most implementations in these cases adopt simple exploration strategies such as $\epsilon$-greedy or softmax (Bellemare et al., 2016).

There are standard model-free techniques to estimate value function in function-approximation scenarios. Because learning $E$-values is simply learning another value-function, the same techniques can be applied for learning $E$-values in these scenarios. In this case, the concept of visit-count – or a generalized visit-count – will depend on the representation of states used by the approximating function.

To test whether $E$-values can serve as generalized visit-counters in the function-approximation case, we used a linear approximation architecture on the MountainCar problem (Moore, 1990) (Appendix

C). To dissociate $Q$ and $E$-values, actions were chosen by an $\epsilon$-greedy agent independently of $E$-values. As shown in Appendix C, $E$-values are an effective way for counting both visits and generalized visits in continuous MDPs. For completeness, we also compared the performance of $LLL$ agents to stochastic agents on a sparse-reward MountainCar problem, and found that $LLL$ agents learns substantially faster than the stochastic agents (Appendix D).

## 4.1 RESULTS – FUNCTION APPROXIMATION

To show our approach scales to complex problems, we used the Freeway Atari 2600 game, which is known as a hard exploration problem (Bellemare et al., 2016). We trained a neural network with two streams to predict the $Q$ and $E$-values. First, we trained the network using standard DQN technique (Mnih et al., 2015), which ignores the E-values. Second, we trained the network while adding an exploration bonus of $\frac{\beta}{\sqrt{-\log E}}$ to the reward (In all reported simulations, $\beta = 0.05$). In both cases, action-selection was performed by an $\epsilon$-greedy rule, as in Bellemare et al. (2016).

Note that the exploration bonus requires $0 < E < 1$. To satisfy this requirement, we applied a logistic activation fucntion on the output of the last layer of the $E$-value stream, and initialized the weights of this layer to 0. As a result, the $E$-values were initialized at 0.5 and satisfied $0 < E < 1$ throughout the training. In comparison, no non-linearity was applied in the last layer of the $Q$-value stream and the weights were randmoly initialized.

We compared our approach to a DQN baseline, as well as to the density model counters suggested by (Bellemare et al., 2016). The baseline used here does not utilize additional enhancements (such as Double DQN and Monte-Carlo return) which were used in (Bellemare et al., 2016). Our results, depicted in Figure 6, demonstrate that the use of $E$-values outperform both DQN and density model counters baselines. In addition, our approach results in better performance than in (Bellemare et al., 2016) (with the mentioned enhancements), converging in approximately $2 \cdot 10^6$ steps, instead of $10 \cdot 10^6$ steps[2].

## 5 RELATED WORK

The idea of using reinforcement-learning techniques to estimate exploration can be traced back to Storck et al. (1995) and Meuleau & Bourgine (1999) who also analyzed propagation of uncertainties and exploration values. These works followed a model-based approach, and did not fully deal with the problem of non-Markovity arising from using exploration bonus as the immediate reward. A related approach was used by Little & Sommer (2014), where exploration was investigated by information-theoretic measures. Such interpretation of exploration can also be found in other works (Schmidhuber (1991); Sun et al. (2011); Houthooft et al. (2016)).

Efficient exploration in model-free RL was also analyzed in PAC-MDP framework, most notably the Delayed $Q$ Learning algorithm by Strehl et al. (2006). For further discussion and comparison of our approach with Delayed $Q$ Learning, see 1.1 and Appendix B.

In terms of generalizing Counter-based methods, there has been some works on using counter-like notions for exploration in continuous MDPs (Nouri & Littman, 2009). A more direct attempt was recently proposed by Bellemare et al. (2016). This generalization provides a way to implement visit counters in large, continuous state and action spaces by using density models. Our generalization is different, as it aims first on generalizing the notion of visit counts themselves, from actual counters to "propagating counters". In addition, our approach does not depend on any estimated model – which might be an advantage in domains for which good density models are not available. Nevertheless, we believe that an interesting future work will be comparing between the approach suggested by Bellemare et al. (2016) and our approach, in particular for the case of $\gamma_E = 0$.

---

[2]We used an existing implementation for DQN and density-model counters available at https://github.com/brendanator/atari-rl. Training with density-model counters was an order of magnitude slower than training with two-streamed network for $E$-values

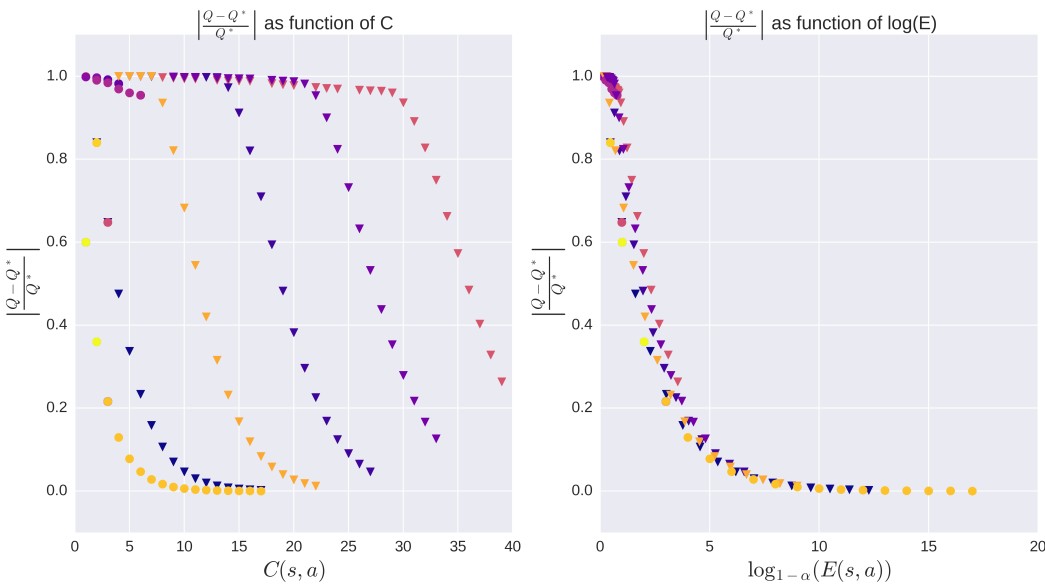

Figure 5: Convergence of $Q$ to $Q^*$ for individual state-action pairs (each denoted by a different color), with respect to counters (left) and generalized counters (right). Results obtained from $E$-Value $LLL$ Softmax on the short bridge environment ($k = 5$). Triangle markers indicate pairs with "east" actions, which constitute the optimal policy of crossing the bridge. Circle markers indicate state-action pairs that are not part of the optimal policy. Generalized counters are a useful measure of the amount of missing knowledge.

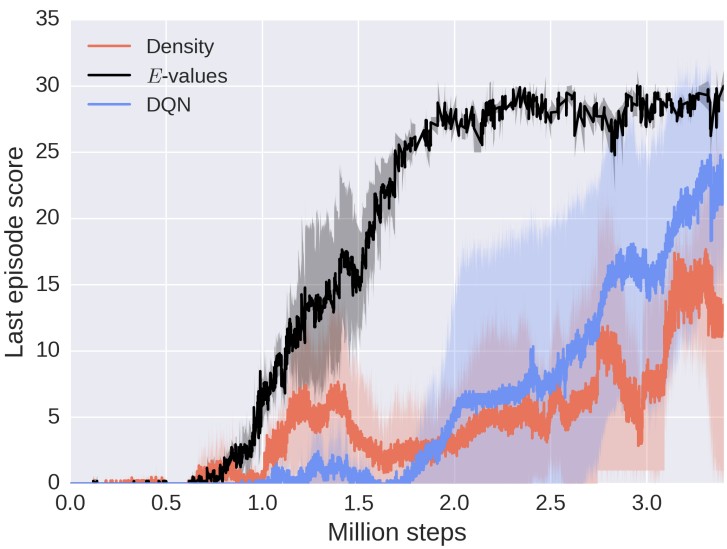

Figure 6: Results on Freeway game. All agents used $\epsilon$-greedy action-selection rule without exploration bonus (DQN, blue), with a bonus term based on density model counters (Density, orange) added to the reward, or with bonus term based on $E$-values (black).

# 6  ACKNOWLEDGMENTS

We thank Nadav Cohen, Leo Joskowicz, Ron Meir, Michal Moshkovitz, and Jeff Rosenschein for discussions. This work was supported by the Israel Science Foundation (Grant No. 757/16) and the Gatsby Charitable Foundation.

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

## A  PROOF OF THE DETERMINIZATION THEOREM

The proof for the determinization mentioned in the paper is achieved based on the following lemmata.

**Lemma A.1.** *The absolute sum of positive and negative differences between the empiric distribution (deterministic frequency) and goal distribution (non-deterministic frequency) is equal.*

$$\sum_{a:f(a)\geq \frac{C(a)}{C}} f(a) - \frac{C(a)}{C} = - \sum_{a:f(a)<\frac{C(a)}{C}} f(a) - \frac{C(a)}{C}$$

*Proof.* Straightforward from the observation that

$$\sum_a f(a) = \sum_a \frac{C(a)}{C} = 1$$

$\square$

**Lemma A.2.** *For any t*

$$\max_a \left\{ \frac{C_t(a)}{t} - f(a) \right\} \leq \frac{1+b(t)}{t}$$

*Proof.* The proof of A.2 is done by induction. For $t = 1$

$$\forall a \in A : \frac{C_t(a)}{t} - f(a) = \max_a \left\{ \frac{C_t(a)}{t} - f(a) \right\}$$

Hence we look at $a \in A$.

$$\frac{C_t(a)}{t} - f(a) \leq \frac{C_t(a)}{t}$$
$$\leq \frac{1+b(1)}{1}$$

assume the claim is true for $t = T$ then for $t = T + 1$ There exists $a$ such that $C_T(a)/T - f(a) \leq b(t)$ which the algorithm chooses for this $a$. For it

$$\frac{C_{T+1}(a)}{T+1} - f(a) = \frac{C_T(a)+1}{T+1} - f(a)$$
$$= \frac{C_T(a)}{T+1} - f(a) + \frac{1}{T+1}$$
$$= \frac{C_T(a) - (T+1)f(a)}{T+1} + \frac{1}{T+1}$$
$$\leq \frac{1+b(t)}{T+1}$$

It also holds that $\forall a' \in A$ s.t. $a' \neq a$

$$\frac{C_{T+1}(a)}{T+1} - f(a) = \frac{C_T(a)}{T+1} - f(a)$$
$$= \frac{C_T(a) - (T+1)f(a)}{T+1}$$
$$< \frac{C_T(a) - Tf(a)}{T+1}$$
$$\leq \frac{1+b(t)}{T+1}$$

$\square$

*Proof of 3.1.* It holds from A.2 together with A.1 that in the step $t$ in the worst case all but one of the actions have $\frac{C_t(a)}{t} - f(a) = \frac{1}{t}$ and the last action has $f(a) - \frac{C_t(a)}{t} = -\frac{|A|-1}{t}$. So by the bound on sum of positives and negatives we get:

$$\lim_{T\to\infty} \frac{C_T(a)}{T} = f(a)$$

$\square$

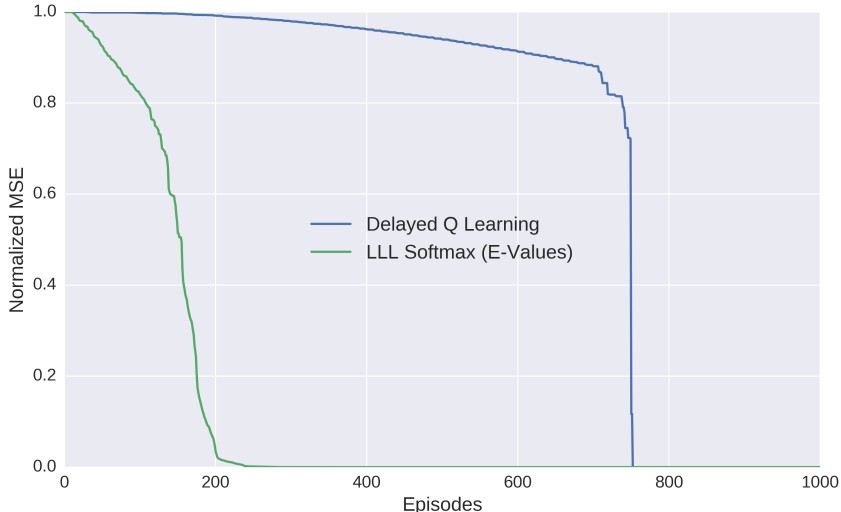

Figure 7: Normalized MSE between $Q$ and $Q^*$ on optimal policy per episode. Convergence of $E$-value $LLL$ and Delayed $Q$-Learning on, normalized bridge environment ($k = 15$). MSE was noramlized for each agent to enable comparison.

## B    COMPARISON WITH DELAYED $Q$-LEARNING

Because Delayed $Q$ learning initializes its values optimistically, which result in a high MSE, we normalized the MSE of the two agents (separately) to enable comparison. Notably, to achieve this performance by the Delayed $Q$ Learning, we had to manually choose a low value for $m$ (in Figure 7, $m = 10$), the hyperparameter regulating the number of visits required before any update. This is an order of magnitude smaller than the theoretical value required for even moderate PAC-requirements in the usual notion of $\epsilon, \delta$, such $m$ also implies learning in orders of magnitudes slower. In fact, for this limit of $m \to 1$ the algorithm is effectively quite similar to a "Vanilla" $Q$-Learning with an optimistic initialization, which is possible due to the assumption made by the algorithm that all rewards are between 0 and 1. In fact, several exploration schemes relying on *optimism in the face of uncertainty* were proposed (Walsh et al., 2009). However, because our approach separate reward values and exploratory values, we are able to use optimism for the latter without assuming any prior knowledge about the first – while still achieving competitive results to an optimistic initialization based on prior knowledge.

## C    EVALUATING $E$-VALUES DYNAMICS IN FUNCTION-APPROXIMATION

To gain insight into the relation between E-values and number of visits, we used the linear-approximation architecture on the MountainCar problem. Note that when using $E$-values, they are generally correlated with visit counts both because visits result in update of the $E$-values through learning and because $E$-values affect visits through the exploration bonus (or action-selection rule). To dissociate the two, $Q$-values and $E$-values were learned in parallel in these simulation, but action-selection was independent of the $E$-values. Rather, actions were chosen by an $\epsilon$-greedy agent. To estimate visit-counts, we recorded the entire set of visited states, and computed the empirical visits histogram by binning the two-dimensional state-space. For each state, its visit counter estimator $\tilde{C}(s)$ is the value of the matching bin in the histogram for this state. In addition, we recorded the learned model (weights vector for $E$-values) and computed the $E$-values map by sampling a state for each bin, and calculating its $E$-values using the model. For simplicity, we consider here the resolution of states alone, summing over all 3 actions for each state. That is, we compare $\tilde{C}(s)$ to $\sum_a \log_{1-\alpha} E(s, a) = C_E(s)$. Figure 8 depicts the empirical visits histogram (left) and the estimated $E$-values for the case of $\gamma_E = 0$ after the complete training. The results of the analysis show that, roughly speaking, those regions in the state space that were more often visited, were also associated with a higher $C_E(s)$.

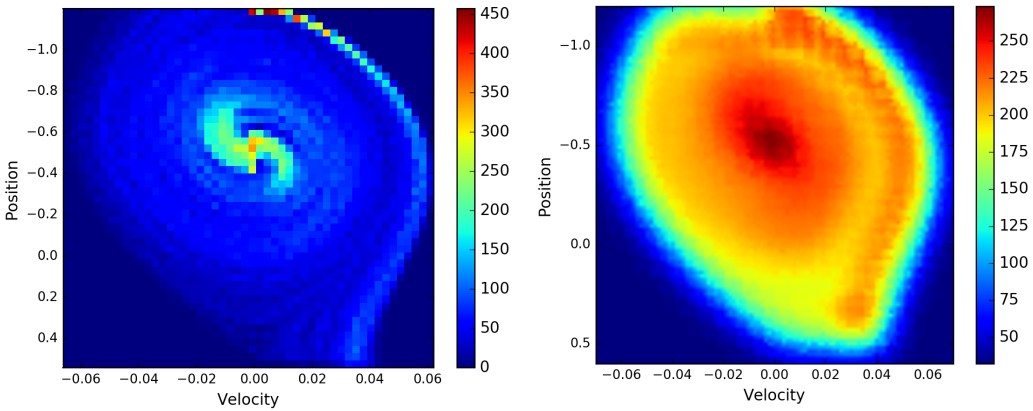

Figure 8: Empirical visits histogram (left) and learned $C_E$ (right) after training, $\gamma_E = 0$.

To better understand these results, we considered smaller time-windows in the learning process. Specifically, Figure 9 depicts the empirical visit histogram (left), and the corresponding $C_E(s)$ (right) in the first 10 episodes, in which visits were more centrally distributed. Figure 10 depicts the *change* in the empirical visit histogram (left), and *change* in the corresponding $C_E(s)$ (right) in the last 10 episodes of the training, in which visits were distributed along a spiral (forming an near-optimal behavior). These results demonstrate high similarity between visit-counts and the $E$-value representation of them, indicating that $E$-values are good proxies of visit counters.

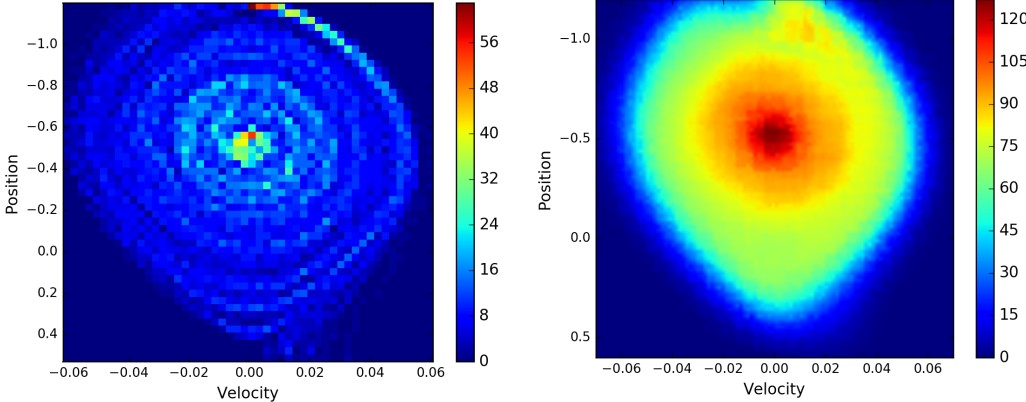

Figure 9: Empirical visits histogram (left) and learned $C_E$ (right) in the first 10 training episodes, $\gamma_E = 0$.

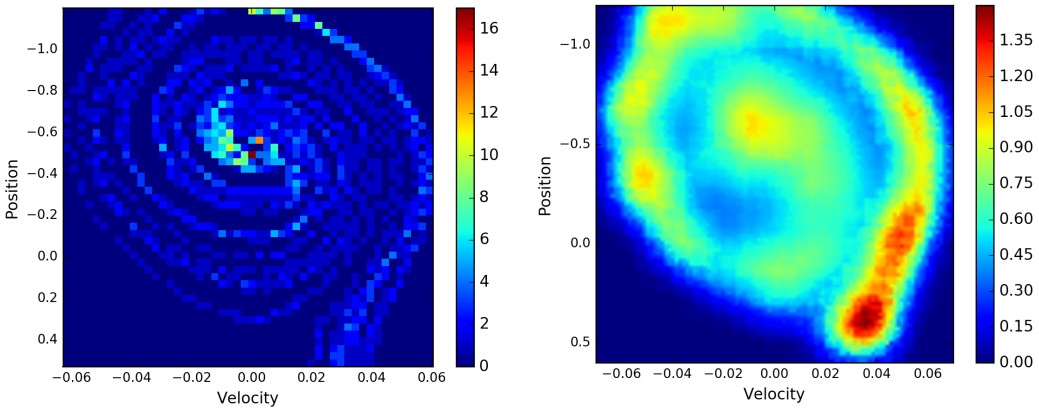

Figure 10: Difference in empirical visits histogram (left) and learned $C_E$ (right) in the last 10 training episodes, $\gamma_E = 0$.

The results depicted in Figures 9 and 10 were achieved with $\gamma_E = 0$. For $\gamma_E > 0$, we expect the generalized counters (represented by $E$-values) to account not for standard visits but for "generalized visits", weighting the trajectories starting in each state. We repeated the analysis of Figure 10 for the case of $\gamma_E = 0.99$. Results, depicted in Figure 11, shows that indeed for terminal or near-terminal states (where position$> 0.5$) generalized visits, measured by difference in their generalized counters, are higher – comparing to far-from terminal states – than the empirical visits of these states (comparing to far-from terminal states).

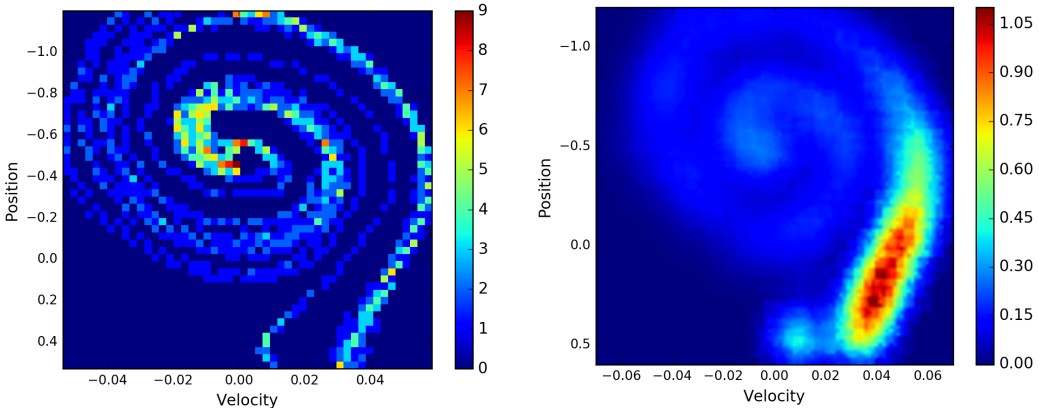

Figure 11: Difference in empirical visits histogram (left) and learned $C_E$ (right) in the last 10 training episodes, $\gamma_E = 0.99$. Note that the results are based on a different simulation than those in Figure 10.

To quantify the relation between visits and $E$-values, we densely sampled the (achievable) state-space to generate many examples of states. For each sampled state, we computed the correlation coefficient between $C_E(s)$ and $\tilde{C}(s)$ throughout the learning process (snapshots taken each 10 episodes). The values $\tilde{C}(s)$ were estimated by the empirical visits histogram (value of the bin corresponding to the sampled state) calculated based on visits history up to each snapshot. Figure 12, depicting the histogram of correlation coefficients between the two measures, demonstrating strong positive correlations between empirical visit-counters and generalized counters represented by $E$-values. These results indicate that $E$-values are an effective way for counting effective visits in continuous MDPs. Note that the number of model parameters used to estimate $E(s,a)$ in this case is much smaller than the size of the table we would have to use in order to track state-action counters in such binning resolution.

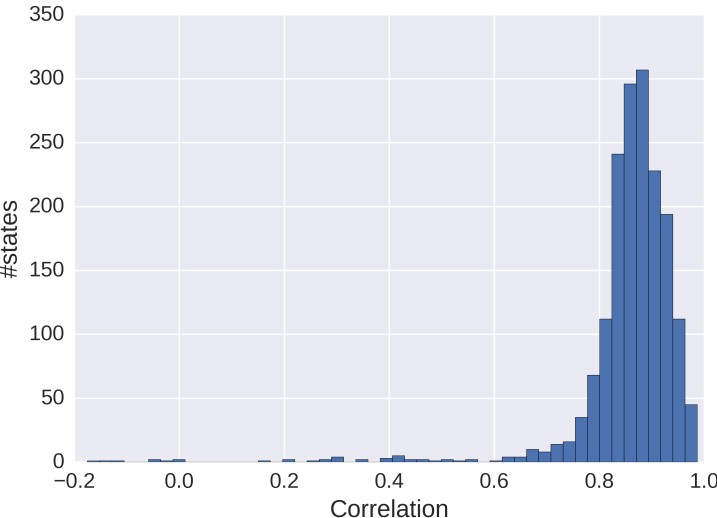

Figure 12: Histogram of correlation coefficients between empirical visit counters and $C_E$ throughout training, per state ($\gamma_E = 0$).

## D  RESULTS ON CONTINUOUS MDPS – MOUNTAINCAR

To test the performance of $E$-values based agents, simulations were performed using the Mountain-Car environment. The version of the problem considered here is with sparse and delayed reward, meaning that there is a constant reward of $0$ unless reaching a goal state which provides a reward of magnitude $1$. Episode length was limited to $1000$ steps. We used linear approximation with tile-coding features (Sutton & Barto, 1998), learning the weights vectors for $Q$ and $E$ in parallel. To guarantee that $E$-values are uniformly initialized and are kept between $0$ and $1$ throughout learning, we initialized the weights vector for $E$-values to $0$ and added a logistic non-linearity to the results of the standard linear approximation. In contrast, the $Q$-values weights vector was initialized at random, and there was no non-linearity. We compared the performance of several agents. The first two used only $Q$-values, with a softmax or an $\epsilon$-greedy action-selection rules. The other two agents are the DORA variants using both $Q$ and $E$ values, following the $LLL$ determinization for softmax either with $\gamma_E = 0$ or with $\gamma_E = 0.99$. Parameters for each agent (temperature and $\epsilon$) were fitted separately to maximize performance. The results depicted in Figure 13 demonstrate that using $E$-values with $\gamma_E > 0$ lead to better performance in the MountainCar problem

In addition we tested our approach using (relatively simple) neural networks. We trained two neural networks in parallel (unlike the two-streams single network used for Atari simulations), for predicting $Q$ and $E$ values. In this architecture, the same technique of $0$ initializing and a logistic non-linearity was applied to the last linear of the $E$-network. Similarly to the linear approximation approach, $E$-values based agents outperform their $\epsilon$-greedy and softmax counterparts (not shown).

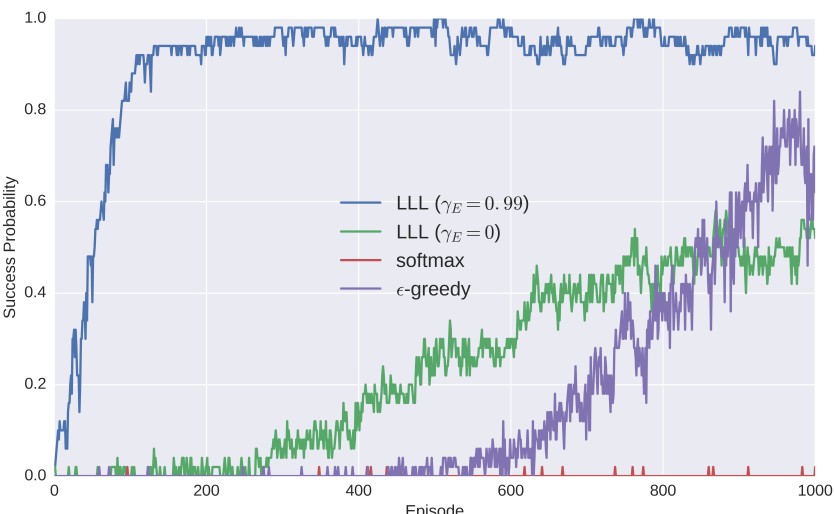

Figure 13: Probability of reaching goal on MountainCar (computed by averaging over 50 simulations of each agent), as a function of training episodes. While Softmax exploration fails to solve the problem within 1000 episodes, $LLL$ $E$-values agents with generalized counters ($\gamma_E > 0$) quickly reach high success rates.

