# OpenReview forum: "DORA The Explorer: Directed Outreaching Reinforcement Action-Selection"
_ICLR.cc/2018/Conference — Accept (Poster)_

### Official Review · AnonReviewer1 · 2017-11-25
**Simple but nice idea to tackle the exploration-exploitation tradeoff in model-free reinforcement learning that seems to work well**

**Rating:** 7
**Confidence:** 3

**Review:**



The paper proposes a novel way for trading of exploration and exploitation in model-free reinforcement learning. The idea is to learn a second (kind of) Q function, which could be called E-function, which captures the value of exploration (E-value). In contrast to the Q-function of the problem at hand, the E-function assumes no preferences among actions.
This is makes sense in my opinion as exploration is exactly “no preferences among actions”.

Actually, to be more precise, the paper shows that the logarithm of E-Values can be thought of as a generalization of visit counters, with propagation
of the values along state-action pairs. This is important, as E-values should actually decrease with repetition. Moreover, the paper shows that by using counters for stochastic action-selection rules commonly employed within the RL community, for every stochastic rule there exist equivalent deterministic rules. Once turned to deterministic counter-based rules, it is again possible improve them using E-values. This provides a nice story for a simple (in a positive sense) approach to tackle the exploration-exploitation tradeoff. The experimental results demonstrate this is a sufficient number of domains. To summarize, for an informed outsider such as the reviewer, the paper makes a simple but strong contribution to an important problem. Overall the paper is well writing and structured.

---

### Official Review · AnonReviewer3 · 2017-11-26
**This paper presents an exploration method for model-free RL that generalizes the counter-based exploration bonus methods and takes into account long term exploratory value of actions rather than a single step look-ahead.**

**Rating:** 6
**Confidence:** 4

**Review:**

This paper presents an exploration method for model-free RL that generalizes the counter-based exploration bonus methods and takes into account long term exploratory value of actions rather than a single step look-ahead. This generalization is achieved by relying on the convergence rate of SARSA updates on an auxiliary MDP.

The method presented in the paper trains a parallel "E-value" MDP, with initial value of 1 for all state-action pairs. It applies SARSA (on-policy) update rule to the E-value MDP, where the acting policy is selected on the original MDP. While the E-value MDP is training, the proposed method uses a 1/log transformation applied to E-values to get the corresponding exploration bonus term for the original MDP. This bonus term is shown to be equivalent counter-based methods for finite MDPs when the discount factor of the E-MDP is set to 0. The paper has minimal theoretical analysis of the proposed algorithm, essentially only showing convergence with infinite visiting. In that regard, the presented method seems like a useful heuristic with anecdotal empirical benefits.

What is crucially lacking from the paper is any reference to model-free Bayesian methods that have very similar intuition behind them: taking into account the long term exploratory benefits of actions (passed on through the Bayesian inference). A comparison would have been trivial to do (with a generic non-informative prior) for the finite MPD setting (section 3.4). Even for the function approximation case one could use Gaussian process methods as the Bayesian baseline. There are also several computationally tractable approximations of Bayesian RL that can be used as baseline for empirical analysis.

It would have also been nice to do some analysis on how the update rule in a function approximation case is affecting the bonus terms. Unlike the finite case, updates to the value of one E-value can change the value for another state-action pair and the convergence could be faster than (1-alpha)^n. Given the lack of any theory on this, an empirical analysis is certainly valuable. (Update: experiment added in the a later revision to study this effect)

Notes:
- The plots are horrible in a print. I had to zoom 400% into the PDF file to be able to read the plots. Please scale them at least by 200% and use a larger font for the legends.

- Add a minimal description of the initial setup for E-value neural network to section 4.1 (i.e. how the initializing is achieved to have a constant value for all state-action pairs as described in the appendix).

* Note: This review and rating has been partially revised with the updates to the paper after the initial comments.

---

> ### Author Response · Authors · 2017-12-16
> **Response to AnonReviewer3**
>
> We thank the reviewer for the insightful comments and would like to respond to the main issues that the reviewer raises. We will address all remaining issues, including scaling the figures and the legend fonts as requested, in the final version.
>
> Regarding the comment about the "anecdotal empirical benefits" of the presented method, please refer to revised Fig. 6, where we demonstrated the benefits of our method in the Freeway Atari 2600 game, which is known as a hard exploration problem (see also response to AnonReviewer 2).
>
> Regarding "reference to model-free Bayesian methods", we are not sure that we correctly understood the comment. The Bayesian approach (e.g, using GPTD) and the use of E-values are not mutually exclusive. While E-values are a proxy measure of uncertainty, the more important point is that they support effective action-selection rules. The question of choosing actions to promote exploration is left undecided in the Bayesian TD algorithms, where practical implementations adopt some stochastic exploration policies, e.g. e-greedy or some close variants (as in Engel et al., Proc. ICML 2003 and Ghavamzadeh et al., Foundations and Trends in Machine Learning, 8, 2015). Put differently, Bayesian methods can become even more powerful by effectively choosing actions. For example, consider the bridge problem discussed in our paper: clearly, any e-greedy exploration scheme on this problem will have very poor performance, assuming no prior knowledge. If actions are only selected based on the expected value (i.e without taking into account the posterior’s variance) than we are basically back in the "regular" Q-learning, in terms of exploration. It will be interesting to compare the uncertainty measured by E-values and the posterior value-function variance. However, this goes beyond the scope of this paper.
>
> We agree with the reviewer that an evaluation of the learned E-values in the function-approximation scenario is an interesting contribution. In response to the reviewer comment, we now address this question in for the MountainCar problem (Appendix D, Figs. 9-13). We show that the logarithm of E-values with gamma_E=0 well approximate the visit counters, and that the logarithm of E-values with gamma_E>0 are a linear function of the visit counters. These results support for the effectiveness of E-values as estimators of visit-counters/generalized visit-counters in continuous MDPs.

---

### Official Review · AnonReviewer2 · 2017-11-27
**E-counters are value functions that must be decayed to zero via learning. Some evidence they help with exploration.**

**Rating:** 6
**Confidence:** 4

**Review:**

The paper proposes an approach to exploration based on initializing a value function to 1 everywhere, then letting the value decay back toward zero as the state space is explored. I like the idea a lot. I don't really like the paper, though. I'd really like to see a strong theoretical and/or empirical justification for it, and both are lacking. On the theoretical side, can a bound be proven for this approach, even in the tabular case? On the empirical side, there are more (and more recent!) testbeds that have come to define the field---the mountain car problem is just not sufficient to convincingly argue that the method scales and generalizes. My intuition is that such an approach ought to be effective, but I really want to see additional evidence. Given the availability of so many RL testbeds, I worry that it had been tried but failed.

Detailed comments:

"Where γ is" -> ", <newline> where γ is".

"The other alternative" -> "The alternative"?

"without learning a model (Mongillo et al., 2014).": Seems like an odd choice for a citation for model-free RL. Perhaps select
the paper that first used the term? Or an RL survey?

Right before Section 1.1, put a period after the Q-learning update equation.

"new states may" -> "new states, may".

"such approaches leads" -> "such approaches lead".

"they still fails" -> "they still fail".

"evaluated with respect only to its immediate outcome": Not so. Several of the cited papers use counters to determine which
states are "known" and then solve an MDP to direct exploration past immediate outcomes.

" exploration bonus(Little & Sommer, 2014)" -> " exploration bonus (Little & Sommer, 2014)".

"n a model-free settings." -> "n model-free settings.".

" Therefore, a satisfying approach for propagating directed exploration in model-free reinforcement learning is still missing. ": I think you should cite http://research.cs.rutgers.edu/~nouri/papers/nips08mre.pdf , which also combines a kind of counter
idea with function approximation to improve exploration.

"initializing E-values to 1": I like this idea. I wonder if one could prove bounds similar to the delayed Q-learning algorithm with
this approach. It is reminiscent of https://arxiv.org/pdf/1205.2606.pdf , which also drives exploration by beginning with an
overly optimistic estimate and letting the data (in a function approximation setting) decay the influence of this initialization.

"So after visited n times" -> "So after being visited n times".

"figure 1a" -> "Figure 1a". (And, in other places.)

"An important property of E-values is that it decreases over repetition" -> "An important property of E-values is that they decrease over repetition".

"t utilize counters, can" -> "t utilize counters can".

" hence we were interested a convergence measure": Multiple problems in this sentence, please fix.

Figure 2: How many states are in this environment? Some description is needed.

Figure 3: The labels in this figure (and all the figures) are absurdly small and, hence, unreadable.

"now turn to show that by using counters," -> "now turn to showing that, by using counters,".

Theorem 3.1: I'm not quite getting why we want to take a stochastic rule and make it deterministic. Note that standard PAC-MDP algorithms choose deterministically. It's not clear why we'd want to start from a stochastic rule.

" models(Bellemare" -> " models (Bellemare".

"Efficient memory-based learning for robot control": This reference is incomplete. (I'm skeptical that it represents the first use of this problem, but I can't check it.)

"Softmax exploration fail" -> "Softmax exploration fails".

"whom also analyzed" -> "who also analyzed".

"non-Markovity" -> "non-Markovianness"?

---

> ### Author Response · Authors · 2017-12-16
> **Response to AnonReviewer2**
>
> We thank the reviewer for the hard work and detailed comments. We will address all of them in the final version, in particular we will scale the figures and legend fonts as requested. Here we address the major comments:
>
> Regarding the comment that "mountain car problem is just not sufficient to convincingly argue that the method scales and generalizes", the MountainCar problem was chosen because of its simplicity, providing a better insight to the algorithm (see also response to AnonReviewer 3). However, we agree with the reviewer that the problem is too simple to convincingly argue that the method scales and generalizes. Therefore, in the revised manuscript (Fig. 6) we tested our approach using the Freeway Atari 2600 game, which is known as a hard exploration problem (Bellemare et al., 2016). Using standard DQN technique (Mnih et al., 2015) without any sophisticated additions or tuning any parameter, we show that the performance of a model that incorporates an E-value exploration bonus exceeds state-of-the-art performance (Bellemare et al., 2016) both in learning speed and in computational efficiency (in fact simulation of the counters suggested by (Bellemare et al., 2016) is computationally so demanding that in the current draft we only show the results up to 2M steps. This simulation is expected to take a few more days to complete).
>
> Regarding the comment that "Several of the cited papers use counters to determine which states are "known" and then solve an MDP to direct exploration past immediate outcomes", to the best of our understanding, they do so in a model-based framework, in which the model parameters are learned. Our approach is unique in being model-free.
>
> Regarding the question of "why one may want to take a stochastic rule and make it deterministic", we agree with the reviewer that in many cases, deterministic algorithms are preferable. Nevertheless, for various reasons, stochastic rules -- in particular epsilon-greedy -- are commonly used in practice. By mapping the stochastic rules to a deterministic rule we show how one can make the minimal change to the stochastic algorithm, which will incorporate the E-values (in the case of gamma_E=0), while preserving exactly the same level of exploration. We consider this point just as an example of utilizing E-values as an exploration bonus.

---

### Public Comment · ~Drew_Davis1 · 2017-12-22
**Reproducibility Challenge**

The Directed Outreaching Reinforcement Action-Selection (DORA) (\cite{Dora}) paper had five primary experiments and we replicated all five. Additionally, we replicated an experiment located in the appendix of the paper to further investigate how E-values compared to optimistic algorithms. For each of these experiments, the authors did not provide code or additional resources beyond the originally submitted paper. Beyond the experiments found in the original submission, we performed two additional experiments under function approximation settings: one in the bridge environment and one in the cart pole environment. We found that tabular environments are reproducible while the function approximation task is brittle. We perform additional experiments and identified misreported hyperparameter as the primary cause of failure in continuous state space.


Full write up and results available at https://github.com/nathanwang000/deep_exploration_with_E_network

(completed by Drew Davis, Jiaxuan Wang, and Tianyang Pan)

---

> ### Author Response · Authors · 2017-12-31
> **Response to Reproducibility Challenge**
>
> We were pleased to see that DORA algorithm was found of sufficient interest for the Reproducibility Challenge, and we would like to thank Drew Davis, Jiaxuan Wang, and Tianyang Pan (DWP) for their effort and comments.
>
> DWP report that while they were able to reproduce our results in the tabular environments, they failed to reproduce the MountainCar function approximation results. In fact, their figures imply that within 3000 episodes, neither LLL nor standard DQN reach satisfying levels of performance.
>
> Going over their code, we find that their failure to reproduce our results (and hence their claim for “misreported hyperparameter”) stems from the fact that DWP attempted to solve a MountainCar problem, in which episode length is 200 steps. By contrast, we followed Sutton and Barto's definition of the MountainCar problem (see fig. 10.2 in the 2nd edition of 'Reinforcement Learning') and used episodes of length 1000 steps. This point will be clarified in the revised manuscript. In addition to this important difference, the temperature parameter used by DWP was T=1 (More precisely, DWP implementation does not explicitly include a temperature parameter). By contrast, as explained in the text, we fitted the temperature parameter to optimize learning (and used T=0.5). This, however, is not an essential point. We tested DWP code on the 1,000-steps episode MountainCar problem and found that the LLL versions of both e-greedy and softmax (for various T values of T=0.1,0.5,1) learn faster than their DQN equivalents.
>
> Another issue raised by DWP is that “DORA’s experiments using function approximation put DQN into a disadvantageous position (not a fair comparison). We are able to adjust the setting to get much better result using DQN.” Simulating the code of DWP, without changing any parameters, we find that the example presented in the modified setting of DWP is not representative. On average, LLL does better than its stochastic counterpart. This is particularly pronounced when considering softmax action-selection. Simulating DWP code, we find that stochastic softmax fails to learn within 3000 episodes whereas its LLL counterpart can learn the task within less than 2000 episodes. Finally, we would like to comment that DWP consider very different settings from the ones we considered. In their simulations, they changed the network architecture, the non-linearity of the activation function, the update frequency of the parameters of the network, the learning rate and more. While we agree that this (modified settings) might be a better baseline (in the Appendix of the revised manuscript we now use a different baseline -- linear approximation with tile coding features), the finding that LLL agents work well in these settings is, in fact, another evidence for the robustness of our approach.
>
> Finally, we would like to note that following the reviewers' comments, the performance of our approach in the function-approximation case is now demonstrated on a larger scale problem, namely FreeWay (see revised manuscript and response to reviewers).
>
> All code will become available on GitHub after the manuscript is accepted, this is to keep the anonymity of the writers.

---

### Author Response · Authors · 2018-01-04
**Revised manuscript**

We thank the reviewers again for their hard work and useful comments, which have significantly improved the manuscript. In the revised manuscript we have addressed all remaining suggestions. Specifically, we scaled the figures, fixed typos and added additional references, as suggested by Reviewer2.

---

### Decision · Program_Chairs · 2018-01-29
**ICLR 2018 Conference Acceptance Decision**

**Decision:**

Accept (Poster)

**Comment:**

This is a very interesting paper that also seems a little underdeveloped. As noted by the reviewers, it would have been nice to see the idea applied to domains requiring function approximation to confirm that it can scale -- the late addition of Freeway results is nice, but Freeway is also by far the simplest exploration problem in the Atari suite. There also seems to be a confusion between methods such as UCB, which explore/exploit, and purely exploitative methods. The case gamma_E > 0 is also less than obvious. Given the theoretical leanings of the paper, I would strongly encourage the authors to focus on deriving an RMax-style bound for their approach.